# Identification of *Trypanosoma cruzi* Growth Inhibitors with Activity In Vivo within a Collection of Licensed Drugs

**DOI:** 10.3390/microorganisms9020406

**Published:** 2021-02-16

**Authors:** Nieves Martinez-Peinado, Nuria Cortes-Serra, Julian Sherman, Ana Rodriguez, Juan M. Bustamante, Joaquim Gascon, Maria-Jesus Pinazo, Julio Alonso-Padilla

**Affiliations:** 1Barcelona Institute for Global Health (ISGlobal), Hospital Clínic—University of Barcelona, 08036 Barcelona, Spain; nieves.martinez@isglobal.org (N.M.-P.); nuria.cortes@isglobal.org (N.C.-S.); quim.gascon@isglobal.org (J.G.); 2Department of Microbiology, New York University School of Medicine, New York, NY 10010, USA; Julian.Sherman@nyulangone.org (J.S.); Ana.RodriguezFernandez@nyulangone.org (A.R.); 3Center for Tropical and Emerging Global Diseases, University of Georgia, Athens, GA 30602, USA; juanbus@uga.edu

**Keywords:** Chagas disease, *Trypanosoma cruzi*, drugs, repositioning, phenotypic assays

## Abstract

Chagas disease, caused by the parasite *Trypanosoma cruzi (T. cruzi)*, affects more than six million people worldwide, with its greatest burden in Latin America. Available treatments present frequent toxicity and variable efficacy at the chronic phase of the infection, when the disease is usually diagnosed. Hence, development of new therapeutic strategies is urgent. Repositioning of licensed drugs stands as an attractive fast-track low-cost approach for the identification of safer and more effective chemotherapies. With this purpose we screened 32 licensed drugs for different indications against *T. cruzi*. We used a primary in vitro assay of Vero cells infection by *T. cruzi*. Five drugs showed potent activity rates against it (IC_50_ < 4 µmol L^−1^), which were also specific (selectivity index >15) with respect to host cells. *T. cruzi* inhibitory activity of four of them was confirmed by a secondary anti-parasitic assay based on NIH-3T3 cells. Then, we assessed toxicity to human HepG2 cells and anti-amastigote specific activity of those drugs progressed. Ultimately, atovaquone-proguanil, miltefosine, and verapamil were tested in a mouse model of acute *T. cruzi* infection. Miltefosine performance in vitro and in vivo encourages further investigating its use against *T. cruzi*.

## 1. Introduction

Chagas disease, caused by the parasite *Trypanosoma cruzi (T. cruzi)*, is a neglected disease that affects more than six million people worldwide [1]. The main burden of the infection is located in the Americas, where the disease is endemic. However, migratory movements and international travel have shifted its epidemiological profile in the last decades, and the disease is nowadays also found in non-endemic countries [1,2].

Clinically, it presents in two phases. First, there is a short acute phase (4–8 weeks long) that is usually asymptomatic and goes unnoticed and undiagnosed. Then the infection evolves to a chronic phase, in which 30% of the chronically infected subjects will develop cardiac and/or digestive symptoms that can lead to death if untreated [3]. 

Two drugs are currently available to treat *T. cruzi* infection: benznidazole (BNZ) and nifurtimox (NFX). Despite providing a good efficacy against the acute stage in newborns, both drugs have variable efficacy during the chronic phase of the infection [1,4]. Moreover, they require long treatment administration regimens that may lead to toxicity and frequent adverse effects, resulting in discontinuation of treatment in 20–30% of patients [5,6,7,8,9]. Thus, there is an urgent need to develop safer, more effective, short-course chemotherapies for reducing the burden of the disease [10].

In the last few years, several efforts have been directed towards the identification of new active compounds against *T. cruzi* [11,12,13]. Among those drug discovery approaches, repositioning of already licensed drugs should also be contemplated, as it stands out as an attractive fast-track and low-cost strategy [14]. Drug repositioning (also named drug repurposing or drug reprofiling) seeks finding new indications for existing drugs. Benefits of this strategy are multiple compared to the discovery and development of new chemical entities, including shortened developmental time and costs, as well as known pharmacological characteristics and safety profiles [14]. Thereby, identification of a licensed drug with potent and specific activity against *T. cruzi* could rapidly advance to clinical trials [14,15,16].

At present, several drug candidates have been evaluated for Chagas disease following this approach [14,16,17,18]. Two of the most promising, the antifungal azoles posaconazole and E1224, were identified in the context of drug repurposing [16]. They represented the first new drugs clinically evaluated for Chagas disease treatment in decades [19], which meant a breakthrough, despite the fact that they did not meet expectations and reported higher rates of therapeutic failure than BNZ in chronically infected subjects [20,21].

With the aim to identify new drugs to advance in the pipeline, in this study we have evaluated the anti-*T. cruzi* activity of 32 commercial drugs by means of state-of-the-art phenotypic assays. These guided the selection of the most active ones for their evaluation in a mouse model of acute *T. cruzi* infection, a first stage to unveil their potential to be repurposed for Chagas disease.

## 2. Materials and Methods

### 2.1. Ethical Statement

Animal studies were approved by the Institutional Animal Care and Use Committee of New York University School of Medicine (protocol #160720; approved 19 July 2019). This protocol adheres to the guidelines of the Association for Assessment and Accreditation of Laboratory Animal Care International (AAALAC).

### 2.2. Drugs

The collection of drugs used is shown in Table 1. Drugs presented as tablets were pulverized and dissolved in dimethyl-sulfoxide (DMSO) as 10 mmol L^−1^ stock solutions. Those already in solution were used directly in the assays at a starting concentration of 0.8 mg mL^−1^ for defibrotide, 150 mg mL^−1^ for meglumine antimoniate, 50% volume:volume (*v*/*v*) for beclometasone dipropionate-clioquinol, and 2.5% (*v*/*v*) for lidocaine. The drugs that contained two active ingredients (artemether-lumefantrine; atovaquone-proguanil; and piperaquine tetraphosphate-dyhidroartemisinin) were dissolved in DMSO by calculating its concentration based on its active ingredient: artemether, proguanil, and piperaquine tetraphosphate, respectively. A pie chart classification of the drugs used in the study per class is provided in Appendix A.

### 2.3. Host Cells Cultures

Vero (African green monkey kidney epithelial cells) [22], LLC-MK2 (Rhesus monkey kidney epithelial cells) [23], NIH-3T3 (Swiss albino mouse embryo fibroblast cells) [23] and HepG2 (human liver epithelial cells) [24] cultures were kept with Dulbecco´s Modified Eagle´s Medium (DMEM) supplemented with 1% penicillin-streptomycin (P-S) and 10% heat inactivated fetal bovine serum (FBS) at 37 °C, 5% CO_2_ and > 95% humidity as described [23,25]. HepG2 cultures were also supplemented with 10% non-essential amino-acids.

### 2.4. Culture of Parasites

*T. cruzi* trypomastigotes from the Tulahuen strain (Discrete Typing Unit (DTU) VI) expressing β-galactosidase, or from the Brazil strain (DTU I) expressing firefly luciferase were maintained using LLC-MK2 or NIH-3T3 cells as hosts in DMEM supplemented with 2% FBS and 1% penicillin-streptomycin-glutamine (P-S-G) as described [25,55]. Between days 5 and 7, medium containing free-swimming trypomastigotes was harvested and centrifuged for 7 min at 2500 rpm as described [12]. Trypomastigotes were allowed to swim out of the pellet for at least 3 h and used to keep the parasite cycle or for the performance of the anti-parasitic assays.

### 2.5. T. cruzi Growth Inhibition Assay 

Vero cells and trypomastigotes from *T. cruzi* Tulahuen expressing β-galactosidase were harvested, centrifuged and re-suspended in DMEM without phenol red supplemented with 1% PSG, 2% FBS, 1 mmol L^−1^ sodium-pyruvate (Na-pyr) and 25 mmol L^−1^ HEPES [25]. Then, both cell types were counted and diluted at a concentration of 1 × 10^6^ cells mL^−1^ each. Trypan blue was used to check Vero cells viability, which had to be >95% to proceed. Vero and trypomastigote cells solutions were mixed 1:1 (*v*/*v*) and 100 μL of that mix was added per well. That volume has 50,000 host cells and 50,000 parasites, i.e., multiplicity of infection (MOI) = 1. Test plates already contained the drugs dispensed at a starting concentration of 100 µmol L^−1^ following a 2-fold dilution pattern down to 0.05 µmol L^−1^. BNZ was used as control of maximal drug growth inhibition in each round, while each plate contained its own negative and positive controls as described [56,57]. After four days at 37 °C, 50 µL of a PBS solution containing 0.25% NP40 and 500 µmol L^−1^ chlorophenol red-β-D-galactoside (CPRG) substrate were added per well [23]. Plates were incubated at 37 °C for 4 h and absorbance read out at 590 nm using an Epoch Gene5 spectrophotometer. All experiments were performed in triplicate.

### 2.6. Anti-Amastigote Specific Assay

Vero cells were seeded in T-175 flasks (5 × 10^6^ cells per flask) in DMEM supplemented with 1% penicillin-streptomycin and 10% FBS and cultured for 24 h. Then, cells were washed once with PBS and free swimming trypomastigotes (1 × 10^7^ trypomastigotes per flask; MOI = 1) in assay medium were added and allowed 18 h to infect [57]. After that, infected cell monolayers were washed with PBS and detached. Cells were counted and diluted to a concentration of 5 × 10^5^ cells per mL, before adding 100 µL per well to test plates already containing the drugs dispensed as described above [57]. In all cases we included BNZ as control drug, and each plate contained its own negative (Vero cells and parasites) and positive controls (Vero cells) [57].

### 2.7. NIH-3T3 Cells-Based Assays

To perform the anti-parasitic assay, culture media of LLC-Mk2 cells infected with *T. cruzi* Tulahuen expressing β-galactosidase [25] was harvested, centrifuged for 7 min at 1237× *g* in order to eliminate the amastigotes, and trypomastigote forms were allowed to swim out of the pellet for at least 3 h at 37 °C. NIH-3T3 cells and trypomastigote solutions were mixed, and 100 μL of the mix was added per well (with 50,000 cells and 50,000 parasites each, MOI = 1) to 96-well plates already containing the drugs dispensed following a dose-response pattern at a starting concentration of 100 µmol L^−1^ in 100 µL per well (50 µmol L^−1^ final concentration in the well). Amphotericin B (Sigma-Aldrich, St. Louis, MO, USA) was used as positive control at a final concentration of 4 µmol L^−1^ [23,55]. Negative and positive controls were carried in every plate. Read out was performed as described above using a Tecan Spectra Mini plate reader. 

To perform the NIH-3T3 cell toxicity assay, NIH-3T3 cells were added (50,000 cells per well) to 96-well plates already containing the drugs. After 4 days, 20 µL of Alamar Blue (Life Technologies, Eugene, OR, USA) was added. Plates were incubated for 6 h at 37 °C and fluorescence was read using a Labsystems Fluoroskan II plate reader (excitation: 544 nm, emission: 590 nm).

### 2.8. Vero and HepG2 Toxicity Assays

Drugs were added to culture-treated 96-well plates at a starting concentration of 800 µmol L^−1^ and 2-fold diluted for a dose-response analysis [56]. Vero and HepG2 cells were detached, centrifuged, and washed. Cell viability was checked with Trypan blue staining. Then, Vero and HepG2 cells were respectively diluted at a concentration of 5 × 10^5^ and 3.2 × 10^5^ cells per mL, before adding 100 µL per well. Each run contained its own negative and positive controls [56]. Plates were incubated at 37 °C for 4 days or 2 days in the case of Vero or HepG2 cells, respectively. Assays were readout by adding 50 µL per well of a PBS solution containing 10% AlamarBlue reagent (ThermoFischer, Waltham, MA, USA) and incubating the plates for another 6 h at 37 °C before recording the fluorescence intensity in a Tecan Infinite M Nano+ reader (excitation: 530 nm, emission: 590 nm). All experiments were performed in triplicate.

### 2.9. T. cruzi In Vivo Inhibition Assay 

Trypomastigotes from transgenic *T. cruzi* Brazil strain (DTU I) expressing firefly luciferase were purified, diluted in PBS, and injected in 8 week old female Balb/c mice (10^6^ trypomastigotes per mouse). Three days after infection (day 4) mice were anesthetized by inhalation of isoflurane and injected with 150 mg per kg of D-Luciferin Potassium-salt (Goldbio, Saint Louis, MO, USA) dissolved in PBS [55]. Mice were imaged 5 to 10 min after injection of luciferin with an IVIS 100 (Xenogen, Alameda, CA, USA), and the data acquisition and analysis were performed with the software LivingImage (Xenogen, Alameda, CA, USA). On the same day 4, treatment started with oral administration of 200 µL of drugs at 30 mg per kg for atovaquone-proguanil (Malarone) and miltefosine (Impavido) [39,45], and 5 mg per kg for verapamil (Manidon) [48]. Drugs were dissolved in PBS vehicle including 0.5% hydroxymethyl cellulose and 0.4% Tween-80. The positive control group received a standard dose of 30 mg per kg of BNZ, while the negative control group received only vehicle solution [55]. Each group was formed by 5 mice. Treatment was continued daily for 10 days. On the scheduled days 4, 9 and 14, corresponding to baseline, 5 and 10 days of treatment, mice were imaged after anesthesia and injection of luciferin as described above.

### 2.10. Data Analysis 

Absorbance and fluorescence values derived from the anti-*T. cruzi* and cell toxicity assays were normalized to the controls [12]. IC_50_ and TC_50_ values are provided as mean and standard deviation (SD). They were determined with GraphPad Prism 7 software (version 7.00, 2016) using a non-linear regression analysis model defined by Equation (1):Y = 100 ÷ ((1 + X^HillSlope) ÷ (IC_50_^HillSlope))(1)

Total flux values recorded in the in vivo experiments are also shown as mean and SD. We used ANOVA with Bonferroni’s correction for multiple comparisons to determine if the differences between groups were statistically significant.

## 3. Results

### 3.1. Anti-T. cruzi Activity on Vero Cells 

*T. cruzi* inhibition was primarily determined by a dose-response phenotypic assay based on Vero cells infected with *T. cruzi*—Tulahuen strain (DTU VI) expressing β-galactosidase [25]. The reference drug BNZ, included as a control of drug growth inhibition in every round, had an average IC_50_ value of 1.93 (0.82) μmol L^−1^, correlating with previous reports [25,56]. In total, we evaluated 32 drugs by the anti-*T. cruzi* phenotypic assay, wherein seven showed high anti-parasitic activity (Figure 1). Miltefosine and nifedipine were the most potent with IC_50_ values in the sub-micromolar range: 0.018 (0.0015) µmol L^−1^ and 0.19 (0.018) µmol L^−1^, respectively (Table 2). Similarly, an IC_50_ value of 0.016 (0.0015) % (*v*/*v*) was reported for lidocaine (Table 2). In addition, atovaquone-proguanil and pentamidine also exhibited an anti-*T. cruzi* activity that exceeded in potency that of the reference drug BNZ, with IC_50_ values of 1.26 (0.14) µmol L^−1^ and 1.01 (0.55) µmol L^−1^, respectively (Table 2). The other two drugs that were considered active (with IC_50_ values within 3× that of BNZ) were verapamil and piperaquine tetraphosphate - dihydroartemisinin, which showed similar IC_50_ values: 3.44 (0.44) µmol L^−1^ and 3.95 (0.51) µmol L^−1^, respectively (Table 2). Chemical structures of these seven drugs are shown in Figure 2. IC_50_ values of the 25 drugs reported as inactive are depicted in Appendix A.

### 3.2. Identification of Drugs with Specific Activity against the Parasite

Drugs that influence host cell viability would also be identified as active (false positive) in the phenotypic anti-parasitic assay. To select active ones with specific activity against *T. cruzi*, we evaluated the seven anti-parasitic active drugs in a secondary cell toxicity assay based on Vero cells [56]. In it, the reference drug BNZ had an average TC_50_ value of 242.2 (13.93) µmol L^−1^ (Appendix A) in agreement with that previously reported [56,58]. Overall, the seven drugs showed low toxicity to Vero cells (Figure 1). Their TC_50_/IC_50_ rates or selectivity index (SI) were > 10 in all cases, a threshold that is usually considered to progress compounds with specific anti-parasitic activity [12]. Verapamil was the drug with lower toxicity to Vero cells, and registered a SI rate of 57.4 (Table 2). Similar TC_50_ values were observed for miltefosine, pentamidine and piperaquine tetraphosphate-dihydroartemisinin (Table 2). Miltefosine had the widest specificity window (SI = 4388.3) (Table 2). On the other side, nifedipine and lidocaine presented SI values of 10.4 and 14.4 respectively, and their toxicity to Vero cells was considerably higher than that retrieved for the others [nifedipine TC_50_ = 1.97 (0.27) µmol L^−1^; lidocaine TC_50_ = 0.23 (0.027) *v*/*v*] (Table 2). Therefore, they were discarded from further progression.

### 3.3. Anti-T. cruzi Growth Inhibition of Active Drugs Confirmed

Host cells used for the anti-*T. cruzi* assay can influence its outcome [59]. In the same manner, operational features such as the laboratory and personnel performing the assays may also bias results [60]. To address these issues we performed confirmatory assays of the five selected drugs in a different laboratory (New York University Anti-infectives Screening Core; [61]). These were based on NIH-3T3 cells as hosts and included an anti-*T. cruzi* assay and a secondary NIH-3T3 cell toxicity assay [23]. The following drugs were evaluated: atovaquone-proguanil, miltefosine, pentamidine, piperaquine tetraphosphate-dihydroartemisinin, and verapamil (Figure 3). These were selected for their specificity (SI > 15) and anti-parasitic activity (IC_50_ < 4 µmol L^−1^) in the assays based on Vero cells (Figure 4). To further progress the drugs, a similar anti-parasitic potency and a SI > 10 over NIH-3T3 cells were required (Figure 4).

Miltefosine showed similar IC_50_ values on both anti-*T. cruzi* assays: IC_50_ = 0.018 µmol L^−1^ on Vero cells, and IC_50_ = 0.037 µmol L^−1^ on NIH-3T3 cells; but it presented a considerably higher toxicity to the latter [TC_50_ = 1.95 (0.57) µmol L^−1^] than to Vero cells [TC_50_ = 78.99 (10.55) µmol L^−1^]. Nonetheless, in both cases its SI window prevailed > 10 and indeed was the highest amongst all the drugs evaluated (Table 2). Atovaquone-proguanil and piperaquine tetraphosphate-dihydroartemisinin exhibited a similar activity against *T. cruzi* on the assay based on Vero cells as well as on that based on NIH-3T3 (Table 2). However, the reported cytotoxicity to NIH-3T3 of the latter [TC_50_ = 27.33 (3.68) µmol L^−1^] involved that its SI value was <10, so it was not considered parasite specific and discarded.

Anti-*T. cruzi* activity of pentamidine and verapamil in the assay based on NIH-3T3 cells was more potent than what we previously reported in the anti-*T. cruzi* assay based on Vero cells (Table 2). In terms of NIH-3T3 cells toxicity, although all three drugs also turned out to be more toxic to these than to Vero cells, their SI rates remained >10 (Table 2).

### 3.4. HepG2 Toxicity Assay

We next determined the toxicity over HepG2 cells of the four active drugs with confirmed activity against *T. cruzi*. Assays based on these cells are used to anticipate potential liver toxicity to human subjects [24,62]. Despite the drugs evaluated are approved for treatment of specific diseases and therefore low toxicity indexes would be expected, we included this assay in our selection cascade. BNZ and digitoxin (DTX) were used as reference drugs and their average TC_50_ values were 229.8 (18.54) µmol L^−1^ and 0.43 (0.14) µmol L^−1^, respectively (Appendix A). All four active drugs had low toxicity to HepG2 cells, with TC_50_ values > 25 µmol L^−1^ in all cases (Figure 5, Table 2). Similarly to what was observed in the Vero cells assay, with an average TC_50_ value of 170.5 (13.14) µmol L^−1^, verapamil had lower toxicity to HepG2 than the other drugs (Table 2). Atovaquone-proguanil and pentamidine had similar toxicities, with TC_50_ values of 34.36 (5.88) µmol L^−1^ and 39.40 (5.20) µmol L^−1^, respectively (Table 2).

### 3.5. Anti-Amastigote Specific Activity of Selected Drugs

Amastigotes are likely the main target for any prospective anti-*T. cruzi* drug. Therefore, we progressed to anti-amastigote experiments those drugs that had been selective on the two mammalian cell lines (Vero and NIH-3T3 cells), had a TC_50_ > 25 μmol L^−1^ against HepG2 cells, and could be administrated orally in an in vivo model, which was the next step of the screening cascade (Figure 4). In each anti-amastigote assay we included the reference drug BNZ, registering an average IC_50_ value of 2.66 (0.14) µmol L^−1^. Atovaquone-proguanil and miltefosine reported high anti-amastigote activity with IC_50_ values more potent than that of BNZ (Table 2, Figure 6). Moreover, both drugs reached SI values > 10 with respect to Vero cells. On the contrary, verapamil did not show specific anti-amastigote activity and had SI < 2 (Table 2, Figure 6).

### 3.6. In Vivo Anti-T. cruzi Activity of the Selected Drugs in a Mouse Model of Acute Infection

We evaluated atovaquone-proguanil, miltefosine, and verapamil in vivo with an acute mouse model of *T. cruzi* infection based on transgenic parasites expressing luciferase (Figure 7) [55]. In this assay, the signal provided by luciferase in the animals is proportional to the *T. cruzi* load, and it is considered a surrogate for parasitemia [55]. One of the mice treated with miltefosine died of unknown causes after the first imaging time point (baseline). A group of animals inoculated with vehicle was included as negative control; other group of mice received a standard dose of BNZ (30 mg/kg/day) as positive control.

We observed that when *T. cruzi*-derived luciferase activity was registered after 5 days of treatment, only verapamil (*p* value = 0.038) and BNZ (*p* = 0.021) showed statistically significant differences versus the vehicle group (Figure 8). Such differences were also statistically significant between animals treated with atovacuone-proguanil (*p* < 0.01) and miltefosine (*p* < 0.05) compared to those treated with BNZ. In all groups except that of animals treated with atovaquone-proguanil, average *T. cruzi* parasitemia after 10 days of treatment had significantly lower levels of parasites compared to vehicle group (Figure 8). Levels of significance of those differences were *p* < 0.01 for miltefosine- and verapamil-treated animals, and *p* < 0.001 for BNZ-treated controls. Although differences between miltefosine and verapamil groups with that of BNZ were non-significant, BNZ-treated mice outperformed the former two by 2 × Log (Figure 8).

## 4. Discussion

Considering the long time and high costs of developing new drugs, repurposing of already licensed ones emerges as a reasonable strategy for the treatment of Neglected Tropical Diseases [14]. Recent assessment for Chagas disease of several anti-fungal azole-derivatives in clinical trials has followed that logic. With that inspiration, we selected 32 commercial drugs and tested them for their anti-*T. cruzi* activity. Seven showed high activity against the parasite, and according to the literature only four of them had been previously evaluated against *T. cruzi*: nifedipine [27], verapamil [46,47,48,49], pentamidine [50,51] and miltefosine [39,40,41,42]. After assessing host cell toxicity over Vero cells, lidocaine and nifedipine were discarded because they returned a higher toxicity rate than the rest. Anti-*T. cruzi* activity of the five remaining drugs was subject to confirmatory analysis by means of an anti-parasitic assay based on NIH-3T3 cells as hosts, which confirmed that atovaquone-proguanil, miltefosine, pentamidine, and verapamil had anti-parasitic IC_50_ values < 4 µmol L^−1^ (i.e., their IC_50_ was within 3× that of BNZ), SI windows to Vero and NIH-3T3 cells were > 10, and HepG2 TC_50_ > 25 µmol L^−1^ [12]. Piperaquine tetraphosphate-dihydroartemisinin was toxic to NIH-3T3 and discarded from further analysis, which highlights the relevance of addressing anti-*T. cruzi* activity in more than one assay [12,59,63,64].

We further assessed the specific anti-amastigote activity of atovaquone-proguanil, miltefosine, and verapamil. Even though it was active and specific against *T. cruzi*, we discarded pentamidine given its history of severely toxic side effects upon parenteral inoculation [65,66]. Moreover, the lack of synergy between BNZ and pentamidine in an in vivo model of *T. cruzi* infection has been reported [51], and it was previously observed that it had lower activity than BNZ [50]. Thus, we decided to test in vivo the same three drugs that went to the anti-amastigote assay. Among them, miltefosine and verapamil showed statistically significant anti-parasitic activity in comparison to vehicle treated animals.

Previous studies have suggested that verapamil could have a cardio-protective effect since it was observed to ameliorate cardiac functions, mortality and morbidity in *T. cruzi* chronically infected mice [46,47]. Verapamil was the first calcium channel antagonist to be introduced into therapy in the early 1960s for the treatment of high blood pressure, heart arrhythmias, and angina [46]. To the best of our knowledge, this is the first time that verapamil is reported to have anti-*T. cruzi* activity in vitro by means of two anti-parasitic assays based on Vero and NIH-3T3 cells. Works by Tanowitz and collaborators exploring the mechanism of action of verapamil and other calcium-channel blockers showed no inhibition of intracellular *T. cruzi* amastigotes in vitro, with which our results would correlate. However, they also reported no differences in parasitemia levels between untreated and verapamil-treated CD1 mice infected with *T. cruzi* parasites of Brazil strain, even though mortality was reduced [48,49]. Contrarily to that, we observed a statistically significant decrease of parasitemia in verapamil-treated animals in comparison to those that received vehicle, which would correlate with the anti-parasitic activity observed in the primary in vitro assay. Parasite and/or mouse strains influence the experimental outcome [59,67], and may explain to some extent the divergence of our verapamil in vivo results from those of previous studies, due to having relied on distinct mouse strains. In any case, the absence of specific anti-amastigote activity of verapamil constitutes a drawback for its progression as anti-*T. cruzi* drug.

Regarding miltefosine, although its mode of action is not fully understood, it is known to inhibit the synthesis of phosphatidylcholine in the cytosol and cytochrome-c oxidase in the mitochondria, which leads to mitochondrial dysfunction and apoptosis-like cell death [68]. A recent work by Pinto-Martinez and collaborators have provided further insight pointing at a double drug-derived action: opening of the sphingosine-activated plasma membrane Ca^+2^ channel, and direct effect on the acidocalcisome, an organelle with osmoregulatory function in *T. cruzi* [69,70]. The combine of both actions will lead to a large parasite-specific intracellular accumulation of Ca^+2^ [69]. Since disruption of the intracellular Ca^+2^ homeostasis has been described as an important drug target in trypanosomatids [40,42,71], miltefosine could indeed be an interesting drug candidate to pursue for Chagas disease. In fact, several studies have reported in vitro susceptibility of *T. cruzi* parasites to miltefosine [40,41,42]. For instance, Luna and coworkers reported a high susceptibility of a panel of *T. cruzi* strains against miltefosine, with an intracellular parasite growth inhibition (IC_50_), assayed on Vero cells, that ranged between 0.082 (0.01) and 0.63 (0.13) µmol L^−1^ [40]; sub-micromolar IC_50_ values with which our results correlate. In this work, in vitro activity of miltefosine outperforms BNZ by 100× and 2× in the primary and anti-amastigote assays based on Vero cells, leading to the widest selectivity indexes calculated (Table 2).

In vivo evaluation of miltefosine was previously shown to reduce parasitemia (<10^5^ parasites mL^−1^ in blood) in Balb/c mice infected with *T. cruzi* Y strain (DTU II) to similar levels as BNZ at day 20 post-infection [42]. Another study described a suppressive activity of miltefosine against both *T. cruzi* Y and Tulahuen strains at a dose of 30 mg per kg administered for 5 days [39]. Inspired by this study, we now provide detailed evidence of miltefosine in vivo anti-parasitic activity against a different *T. cruzi* strain (Brazil) from the widespread and pathogenic DTU I. We have used an infection model based on bioluminescent parasites that provides higher sensitivity than microscopy-based quantification of blood parasitemia.

Originally developed in the 1980s as an anti-cancer agent, miltefosine is the only oral treatment approved for leishmaniasis [68,72]. Its registration to treat fatal visceral leishmaniasis was considered a breakthrough, but some limitations have been observed thereafter, including frequent side effects driving treatment discontinuation and failure, appearance of resistances, and low availability and affordability [68,73]. These features might discourage its use for Chagas disease. However, given the very good results we have observed in vitro, particularly the very wide SI, and its inhibitory capacity in an acute in vivo infection, it is worth the further assessment of miltefosine’s performance in a chronic model of *T. cruzi* infection. Similarly, it remains to be elucidated whether its co-administration with BNZ would serve to reduce the dosage of the latter without compromising its efficacy, with the aim of limiting side effects.

## 5. Conclusions

Despite the recent failure of azole-derivatives in clinical trials, drug repurposing is still the fastest and cheapest strategy to identify new treatments for Chagas disease. Out of the 32 drugs for different indications evaluated in this study, we found that miltefosine and verapamil reduced parasitemia in a mouse model of acute *T. cruzi* infection after ten days of treatment with 30 and 5 mg per kg, respectively, expanding previously obtained results by others. Although a similar dose of BNZ had a higher inhibitory effect, taking into consideration that miltefosine was shown to specifically inhibit the parasite intracellular replicative stage, and its indication for the oral treatment of leishmaniasis, its potential use against *T. cruzi* should be further investigated. Moreover, miltefosine had the widest activity windows, which may allow testing it at higher doses or under alternative regimes, including its co-administration with BNZ.

## Figures and Tables

**Figure 1 microorganisms-09-00406-f001:**
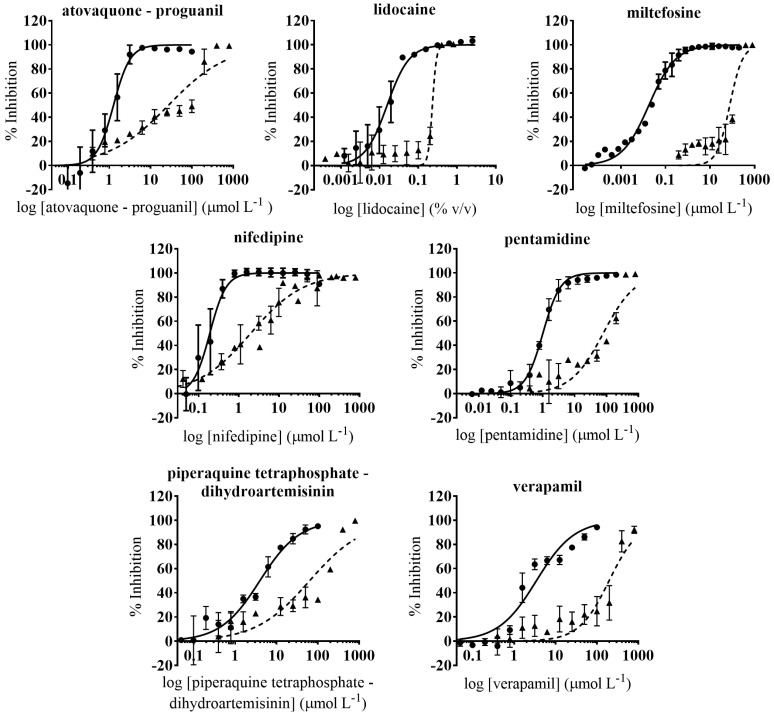
Dose-response curves of anti-*T. cruzi* and Vero cell toxicity assays of the seven drugs that showed an anti-parasitic activity with IC_50_ values < 4 μmol L^−1^. Anti-*T. cruzi* assay data are represented by circles and straight lines, while Vero cells toxicity data are represented by triangles and dashed lines. Graphs represent mean values and SD of at least three independent replicas.

**Figure 2 microorganisms-09-00406-f002:**
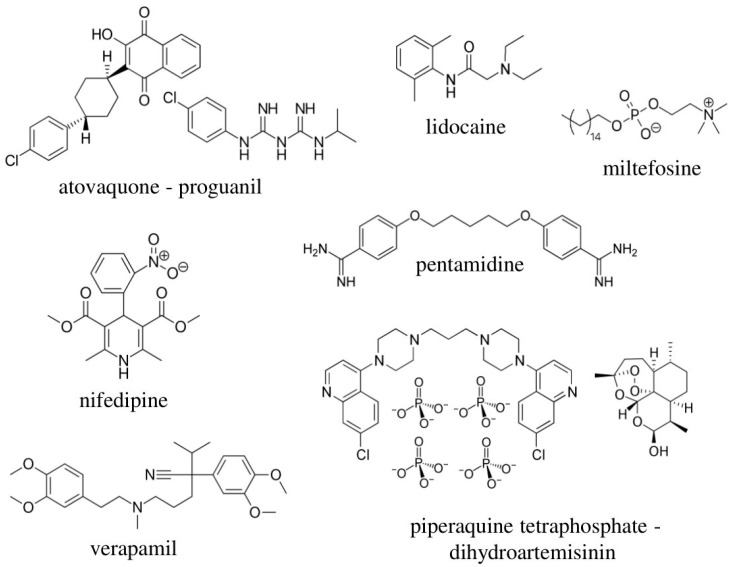
Chemical structures of the seven active drugs identified in the anti-parasitic assay based on Vero cells as hosts.

**Figure 3 microorganisms-09-00406-f003:**
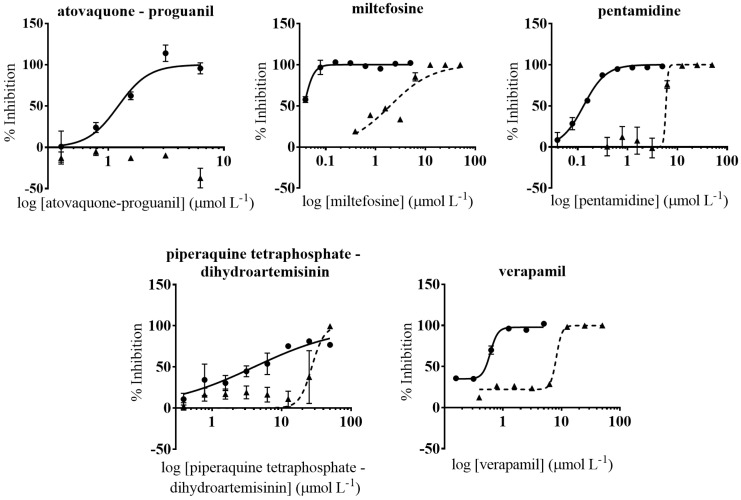
Dose-response curves of anti-*T. cruzi* and cell toxicity assays based on NIH-3T3 cells. Anti-*T. cruzi* assay data are represented by circles and straight lines, while NIH-3T3 cells toxicity assay data are represented by triangles and dashed lines. Graphs represent mean values and SD of at least three replicas.

**Figure 4 microorganisms-09-00406-f004:**
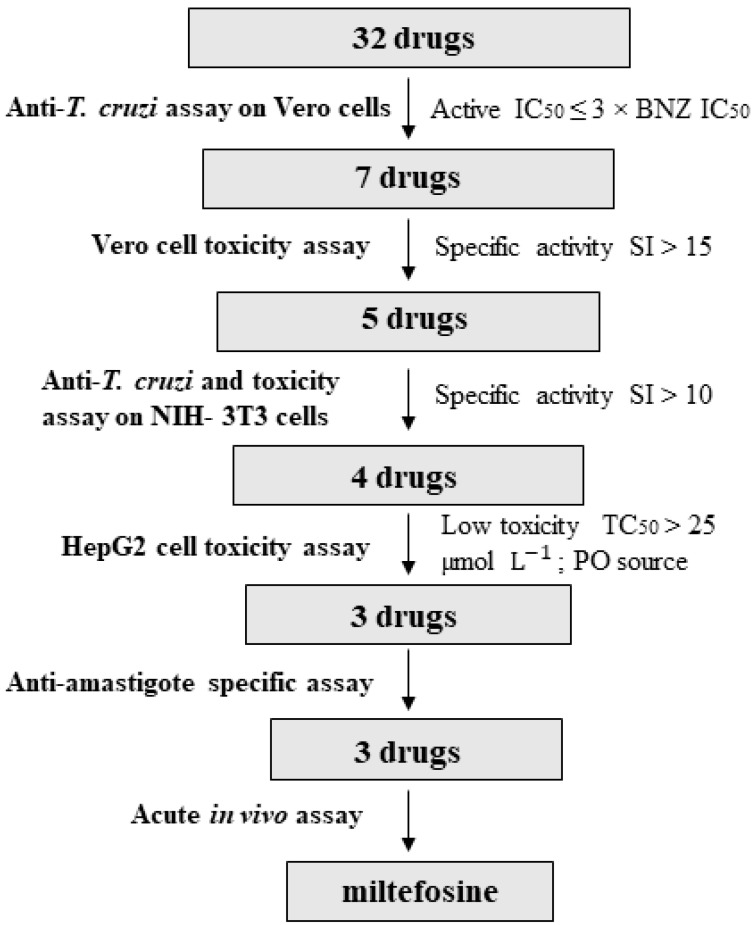
Flow-chart depicting the study drug progression steps (SI, selectivity index; PO, per oral source).

**Figure 5 microorganisms-09-00406-f005:**
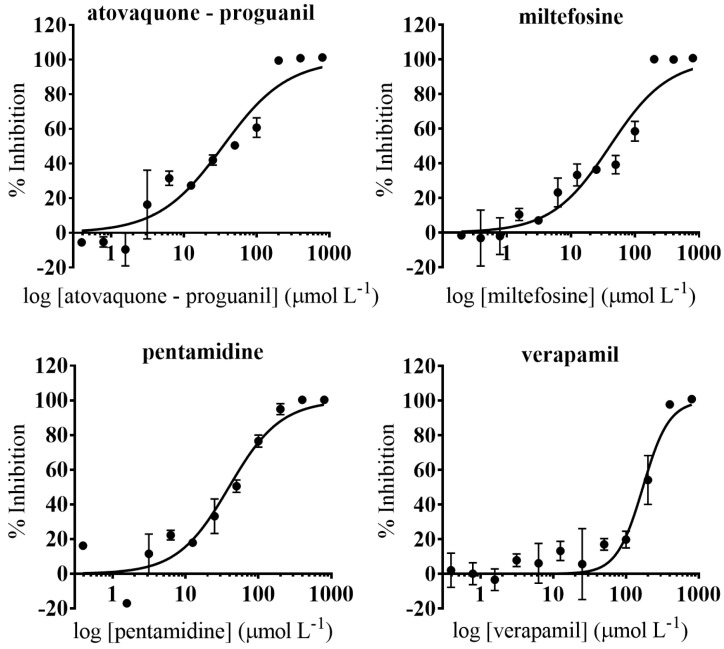
Dose-response curves of the four selected drugs with confirmed anti-*T. cruzi* activity assessed in HepG2 cell toxicity assays. Graphs represent mean values and SD results of at least three independent replicas.

**Figure 6 microorganisms-09-00406-f006:**
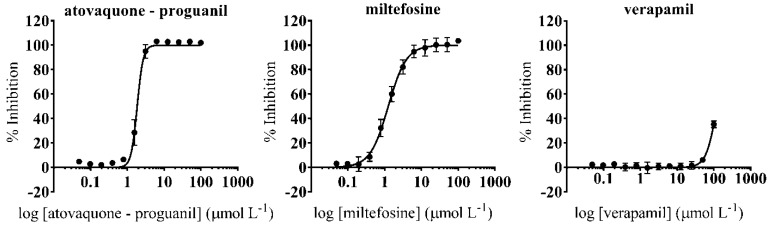
Dose-response curves of progressed drugs that can be orally administered in the anti-amastigote assay.

**Figure 7 microorganisms-09-00406-f007:**
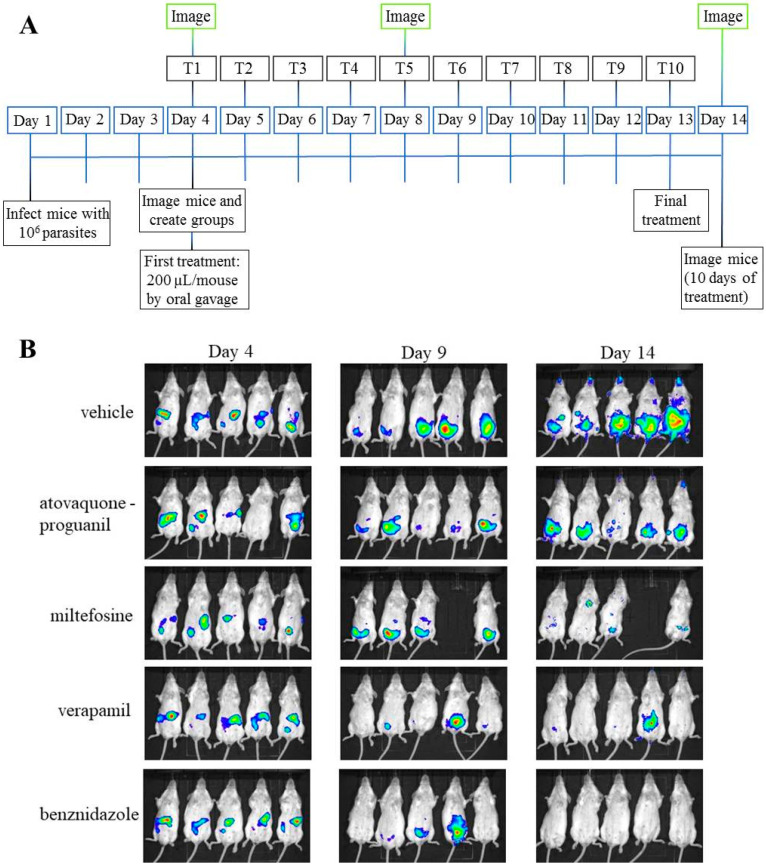
Scheme of the in vivo assessment of the three selected drugs against *T. cruzi* in an acute infection model (**A**) Timeline of events in the acute in vivo study (T1 to T10 depict the days of treatment). (**B**) Images of mice after 0, 5 and 10 days of treatment.

**Figure 8 microorganisms-09-00406-f008:**
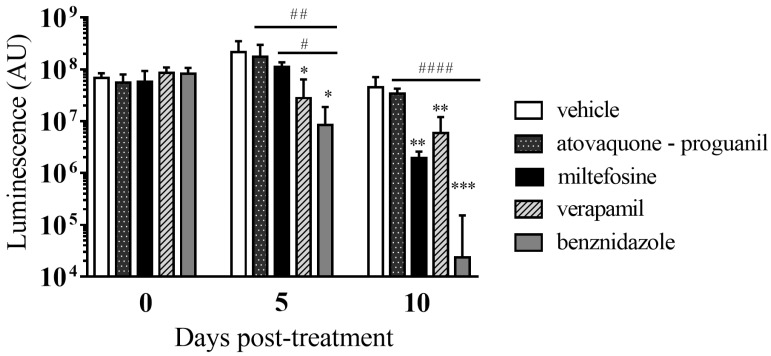
In vivo *T. cruzi* parasitemia (total flux) at baseline (day 0), and after 5 and 10 days of treatment. Statistically significant differences between drug-treated groups versus vehicle group or BNZ-treated group are respectively indicated by * or # symbols. One symbol indicates *p* < 0.05, two indicate *p* < 0.01, three indicate *p* < 0.001, and four symbols indicate *p* < 0.0001.

**Table 1 microorganisms-09-00406-t001:** Drugs used in the study. They are sorted by their “Trade name” in alphabetical order. (Table 1 continues in next page).

Trade Name	Active Ingredient	Class	Route of Administration	Molecular Weight (g/mol)	References
Adalat	nifedipine	calcium channel blocker	oral	346.34	[26,27]
Amlodipine Normon	amlodipine	calcium channel blocker	oral	408.88	[14,26,28]
Apocard	flecainide	antiarrhythmic	oral	414.34	[29]
Atenolol Normon	atenolol	beta blocker	oral	266.34	[14]
Biocoryl	procainamide	antiarrhythmic	oral	235.33	[14]
Bisoprolol Normon	bisoprolol	beta blocker	oral	325.44	[30]
Daraprim	pyrimethamine	antiprotozoal	oral	248.71	[14]
Defitelio	defibrotide	antithrombotic	intravenous	444.40	[31]
Diamicron	gliclazide	antidiabetic	oral	323.41	[32]
Enalapril Normon	enalapril	ACE inhibitor	oral	376.45	[14,33]
Eskazole	albendazole	antihelminthic and antiprotozoal	oral	256.33	[14]
Eurartesim	piperaquine tetraphosphate-dihydroartemisinin	antimalarial	oral	927.49–284.35	[34]
Glucantime	meglumine antimoniate	antileishmanial	intramuscular	365.98	[26,35]
Glucophage	metformin hydrochloride	antidiabetic	oral	129.16	[14]
Hemovas	pentoxifylline	hemorrheologic agent	oral	278.31	[14,36,37]
Humatin	paramomycin	antimicrobial	oral	615.63	[38]
Impavido	miltefosine	antiprotozoal	oral	407.57	[39,40,41,42]
Iver P (ELEA)	ivermectin	antihelminthic	oral	875.10	[43]
Quinine sulfate (Hospital Clinic)	quinine sulfate	antimalarial	oral	782.96	[44]
Lidocaine Braun	lidocaine	local anesthetic	intravenous	234.34	[14]
Lomper	mebendazole	antihelminthic	oral	295.29	[14]
Malarone	atovaquone-proguanil	antimalarial	oral	366.84	[45]
Manidon	verapamil	calcium channel blocker	oral/intravenous	454.60	[46,47,48,49]
Masdil	diltiazem hydrochloride	calcium channel blocker	oral	414.52	[14]
Menaderm Otológico	beclometasone dipropionate-clioquinol	fungal/antibacterial	otic	521.04–304.91	-
Nerdipina	nicardipine	calcium channel blocker	oral	479.53	[14,26]
Pentacarinat	pentamidine	antiprotozoal	intravenous/intramuscular/inhalation	340.42	[14,26,50,51]
Primaquine (Hospital Clinic)	primaquine	antimalarial	oral	259.35	[14,52]
Riamet	artemether-lumefrantine	antimalarial	oral	298.37–528.94	[53]
Solgol	nadolol	beta blocker	oral	309.40	[14]
Sotapor	sotalol	beta blocker	oral	272.36	[54]
Tricolam	tinidazole	antiprotozoal	oral	247.27	[14]

**Table 2 microorganisms-09-00406-t002:** Average IC_50_, TC_50_ and SI values of the seven primary active drugs. These were tested in phenotypic assays based on Vero, NIH-3T3 and HepG2 cells.

Drug	Vero Cells Assays			NIH-3T3 Cells Assays			HepG2 Assay	Anti-Amastigote Assay	
	[IC_50_ (SD)] µmol L^−1^	[TC_50_ (SD)] µmol L^−1^	SI	[IC_50_ (SD)] µmol L^−1^	[TC_50_ (SD)] µmol L^−1^	SI	[TC_50_ (SD)] µmol L^−1^	[IC_50_ (SD)] µmol L^−1^	SI
Benznidazole	1.93 (0.82)	242.2 (13.93)	125.5	-	-	-	229.8 (18.54)	2.66 (0.14)	91.1
Atovaquone-proguanil	1.26 (0.14)	27.13 (5.05)	21.5	1.32 (0.07)	>50	>50	34.36 (5.88)	1.85 (0.06)	14.7
Miltefosine	0.018 (0.0015)	78.99 (10.55)	4388.3	0.037 (0.001)	1.95 (0.57)	52.7	51.28 (7.51)	1.25 (0.05)	63.2
Lidocaine ^#^	0.016 (0.0015)	0.23 (0.027)	14.4	-	-	-	-	-	-
Nifedipine	0.19 (0.018)	1.97 (0.267)	10.4	-	-	-	-	-	-
Pentamidine	1.01 (0.55)	78.96 (15.55)	78.2	0.13 (0.005)	5.9 (0.15)	45.4	39.4 (5.20)	-	-
Piperaquine tetraphosphate-dihydroartemisinin	3.95 (0.51)	75.27 (16.56)	19.1	4.05 (0.72)	27.33 (3.68)	6.8	-	-	-
Verapamil	3.44 (0.44)	197.4 (25.54)	57.4	0.60 (0.04)	8.16 (1.63)	13.6	170.5 (13.14)	122.5 (7.04)	1.6

^#^ Lidocaine values refer to percentage of drug volume per well considering its dose-response distribution in the assay plates started at a concentration of 2.5% (*v*/*v*). SD, standard deviation; SI, selectivity index.

## Data Availability

The data presented in this study are contained within the article or Appendix A.

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
