# Peer review of "Identification of Trypanosoma cruzi Growth Inhibitors with Activity In Vivo within a Collection of Licensed Drugs"

_microorganisms, 2021, doi:10.3390/microorganisms9020406_

Round 1

Reviewer 1 Report

Trypanosoma cruzi, responsible for Chagas disease, is a serious public health problem in Latin America with even seven-eight million people infected in the world. Diagnosis of T. cruzi infection is difficult, and the treatment is usually effective only in the acute stage; patients with chronic phase of Chagas disease mostly receive only palliative care. Therefore, the research conducted by Martines-Peinado N, Cortes-Serra N, Sherman J, Rodriguez A, Bustamante JM, Gascon J, Pinazo M-J, and Alonso-Padilla J is very important and influencial in the field. The study was very interesting and it was a pleasure to read and review this paper. Therefore, after editor approval, I consider the present work to be published in Microorganisms.

I have only a few suggestions:

1) Please write "T. cruzi" in italics in the Results section- line 89,90,93,98,106,115,129,133/134,140,145,150,152,156,163,178,193,195,201,203,206,212

2) Please write "in vivo" in italics in the Results section- line 180,192,201,202,212

3) In figure 4, please write "in vivo" in italics

4) I would suggest adding Conclusion section, to summarize the most important findings of the study.

5) In supplementary material: the descriptions of the figure S1 and S2 should be under the figures, not above

Author Response

We thank reviewer 1 for her/his comments. Please find our answers typed in blue below.

1) Please write "T. cruzi" in italics in the Results section- line 89,90,93,98,106,115,129,133/134,140,145,150,152,156,163,178,193,195,201,203,206,212. The term has been changed as suggested.

2) Please write "in vivo" in italics in the Results section- line 180,192,201,202,212. The term has been changed as suggested.

3) In figure 4, please write "in vivo" in italics. The term has been changed within the figure as suggested.

4) I would suggest adding Conclusion section, to summarize the most important findings of the study. A new “Conclusions” section has now been included.

5) In supplementary material: the descriptions of the figure S1 and S2 should be under the figures, not above. Descriptors have now been placed under their corresponding supplementary figure.

Reviewer 2 Report

 In this study, the authors screened 32 licensed drugs against T. cruzi. Five drugs showed 22 potent activity rates against it, which were also specific  with respect to host cells. T. cruzi inhibitory activity of four of them was confirmed by a secondary anti-parasitic assay based on NIH-3T3 cells. Then,  assessed their toxicity to human HepG2 cells and anti-amastigote specific activity of those drugs. Atovaquone – proguanil, miltefosine and verapamil were tested in a mouse model of acute T. cruzi infection. However, the manuscript is well written and the data is well presented, there still some comments that need to be addressed

  • You did not describe the dose-response assay in the methods section. You also need to expand its results. 
  • Please add a footnote under each table including the full name of the included abbreviations.
  • Italicize (in vivo and in vitro) through the manuscript. 
  • Please expand the explanation of all the figures in legends.
  • What T1-10 mean?
  • Please add the p-value in the respective places of the result section.
  • Please add the conclusion of your study.
  • it is better to use the term (positive control) instead of (vehicle) throughout the manuscript and the figures.

Author Response

We thank reviewer 2 for her/his comments. Please find our answers below typed in blue color.

  • You did not describe the dose-response assay in the methods section. You also need to expand its results. This has now been further explained in sections 2.5, 2.6, 2.7 and 2.8.
  • Please add a footnote under each table including the full name of the included abbreviations. Abbreviations have been detailed in the Tables footnotes, or directly expanded in the Tables.
  • Italicize (in vivo and in vitro) through the manuscript. These terms have now been italicized. 
  • Please expand the explanation of all the figures in legends. Figures legends have now been expanded with further explanations.
  • What T1-10 mean? T1 to T10 in figure 7 stand for day of treatment. So T1 would be one day of treatment and T10 would be 10 days of treatment. This is now included in the footnote of the figure.
  • Please add the p-value in the respective places of the result section. This has now been included where corresponding in the text (see last paragraph of section 3.6).
  • Please add the conclusion of your study. A new “Conclusions” section has now been included in the revised manuscript.
  • It is better to use the term (positive control) instead of (vehicle) throughout the manuscript and the figures. Vehicle-treated animals conform negative control group, i.e. that one where no drug is provided to potentially interfere with the parasite infection. This is detailed in the text in the first paragraph of section 3.6.